# Analog Gradient Calculation of Optical Activation Function Material

**Jakub Kostial**
Optical Networks
Electronic and Electrical Engineering
University College London
ucapajk@ucl.ac.uk

**Filipe M. Ferreira**
Optical Networks
Electronic and Electrical Engineering
University College London
f.ferreira@ucl.ac.uk

## Abstract

Most realisations of Optical Neural Networks are aimed at using the platform for inference. In backpropagation, the update process is performed in opposite direction to the forward pass and relies on the gradient of the loss function with respect to the weights. To calculate this, the gradient of the activation function with respect to the inputs is required. As such, the training process of Optical Neural Networks is typically implemented in the digital domain. This is realised using simulation techniques however this method is not optimal as it is not fast and cannot replicate the subtle experimental imperfections of the system. One mitigation involves training with additional noise but this is suboptimal. We propose a novel method to implement backpropagation through nonlinear optical units using small-signal modulation of laser inputs and a lock-in amplifier. This allows for the calculation of the gradient of the activation function with respect to the inputs synchronously with the forward pass using high-speed analog circuitry. We experimentally demonstrate this method for a semiconductor optical amplifier in the nonlinear regime and show that the measured gradient is in good agreement with the calculated gradient from a steady-state analytical model for the device. This method can extract the phase of the signal enabling the encoding of negative and complex weights - besides being applicable to any optical nonlinear material, electro-optic activation functions in free space or photonic integrated platforms. Importantly, this gradient measurement method is resilient to device drift, induced by environment or ageing, which would affect finite difference based techniques - those would require periodic calibration.

## 1   Introduction

Backpropagation is regarded as the algorithm of choice for training neural networks. The algorithm is based on recursive application of the chain rule and calculates the gradient of the loss function with respect to the weights [1]. To find the gradients, the information is passed backwards through the network. In the forward pass, the activation function is applied to the input and provides nonlinear modulation. Instead, in the backward pass, the gradient of the activation function with respect to the inputs is required. In Optical Neural Networks (ONNs), it is not trivial to access the gradient as the optical computational steps for the forward pass and backward pass are different. And, thus, it is one of the main challenges in implementing ONNs.

Ideally, the backward pass of the network should be implemented wholly in the optical domain to avoid the need for optical to digital conversion. But finding an optical material that implements this response is nontrivial. Our proposal here it to use an analog opto-electronic method using small-signal modulation and a lock-in amplifier. The benefit of using a lock-in amplifier is that it can

38th Second Workshop on Machine Learning with New Compute Paradigms at NeurIPS 2024(MLNCP 2024).

provide a high speed ($>$10GHz) and scalable platform, being suitable for implementation with photonic integrated circuits. This allows for the extraction of the gradient of the activation function with respect to the inputs synchronously with the forward pass as the weights are updated. Potentially, the output signals can then be used to update the weights of the network with minimal latency [2]. By computing the gradient simultaneously with the forward pass, the need for implementing backpropagation in the digital domain is removed.

Other solutions involve changes to the training algorithm such that backpropagation is avoided or modified and direct access/calculation of the gradients in the optical domain is no longer required. These methods can be neuro-inspired despite taking longer to descend on the optimal solution such as the direct feedback alignment method [3]. Or stochastic perturbation methods where weights are randomly perturbed and the change in the loss function was used to approximate the gradient such as Bandyopadhyay et al. [4]. These methods are not optimal as the convergence is slower compared to backpropagation as discussed in [5].

An experimental demonstration of the backpropagation algorithm, calculating the gradient of the loss function with respect to the weights, by interfering the forward inference and backward error signals was performed by Pai et al. [6]. The authors demonstrated the technique through a MZI based ONN and the readout was then performed using a camera of the tapped outputs at the reference mid section of the circuit. However the authors used activation functions in the digital domain. To implement backpropagation through nonlinear optical materials, Spall et al. [7] utilised the pump probe process with rubidium gas. This method involved building a free space ONN with both forward pump and backward probe propagation components. The gas cell is a type of saturable absorber where the transmissivity is nonlinear with respect to the input power, thus the forward pump induces a nonlinear response on the gas encoding the forward pass For low input power, the transmissivity is low and as the pump power increases (from a threshold point) the transmissivity starts to increase linearly. Both the pump and probe components propagate through the gas cell where the probe is used to measure the gradient of the activation function with respect to the inputs. However, the probe gradient measurement is accurate for a narrow range of pump powers.

One of the most convenient platforms for implementing optical activation functions are saturable absorbers and semiconductor optical amplifiers (SOAs). They have a nonlinear response to the input power and can be both implemented in integrated and free space platforms. Specifically semiconductor saturable mirrors (SESAMs) and SOAs are based on semiconductor materials and their responses are well understood and modelled. The SOA amplifies nonlinearly in accordance with the gain saturation equation [8], for steady-state conditions, in accordance to:

$$\mathbf{g} = \frac{\mathbf{g_0}}{1 + I/I_{sat}} \tag{1}$$

where $\mathbf{g}$ is the gain, $\mathbf{g_0}$ is the small signal gain, $I$ is the input power and $I_{sat}$ is the saturation power. SESAMs reflect nonlinearly in accordance with the saturable absorber equation [8]:

$$\mathbf{a} = \frac{\mathbf{a_0}}{1 + I/I_{sat}} \tag{2}$$

where $\mathbf{a}$ is the absorption and $\mathbf{a_0}$ is the small signal absorption.

## 2    Backpropagation: the relevance of the activation function gradient

To explain backpropagation through nonlinear units we describe a simple 2 layer neural network. The input vector is given by $\mathbf{x}$, the weight matrix of the first layer is given by $\mathbf{W_1}$. By performing a matrix vector multiplication, the output of the first layer is given by $\mathbf{z_1}$.

$$\mathbf{z_1} = \mathbf{W_1}\mathbf{x_1} \tag{3}$$

The output of the first layer is then passed through the activation function to give the input of the second layer $\mathbf{x_2}$.

$$\mathbf{x_2} = f(\mathbf{z_1}) \tag{4}$$

The weight matrix of the second layer is given by $\mathbf{W_2}$ and the output of the second layer is given by $\mathbf{y_2}$.

$$\mathbf{y_2} = \mathbf{W_2}\mathbf{x_2} \tag{5}$$

The loss function is given by $\mathbf{L}$ and the gradient of the loss function with respect to the first weight matrix is calculated by the chain rule. The gradient allows for the update of the weights in order to minimise the loss function. Applying the chain rule, the gradient of the loss function with respect to the first weight matrix is given by:

$$\frac{\partial \mathbf{L}}{\partial \mathbf{W_1}} = \frac{\partial \mathbf{L}}{\partial \mathbf{y_2}} \frac{\partial \mathbf{y_2}}{\partial \mathbf{x_2}} \frac{\partial \mathbf{x_2}}{\partial \mathbf{z_1}} \frac{\partial \mathbf{z_1}}{\partial \mathbf{W_1}} \tag{6}$$

This is rewritten as:

$$\frac{\partial \mathbf{L}}{\partial \mathbf{W_1}} = \frac{\partial \mathbf{L}}{\partial \mathbf{y_2}} \mathbf{W_2} f'(\mathbf{z_1})\mathbf{x_1} \tag{7}$$

Where the first value $\frac{\partial \mathbf{L}}{\partial \mathbf{y_2}}$ is known as the error vector and $f'(\mathbf{z_1})$ is the derivative of the activation function with respect to the input. For a ReLU activation function applied to an example weight matrix the transformation is shown as:

$$ReLU(\mathbf{W}) = ReLU \begin{pmatrix} 0.5 & 1.9 & 0 \\ -0.6 & 0 & -0.2 \\ -1 & 0 & 0.2 \end{pmatrix} = \begin{pmatrix} 0.5 & 1.9 & 0 \\ 0 & 0 & 0 \\ 0 & 0 & 0.2 \end{pmatrix}$$

The derivative of the ReLU activation function is a step function with value 0 for negative inputs and 1 for positive inputs. Hence, the derivative of the ReLU activation function applied to the example weight matrix is:

$$f'(\mathbf{W}) = ReLU'(\mathbf{W}) = \begin{pmatrix} 1 & 1 & 0 \\ 0 & 0 & 0 \\ 0 & 0 & 1 \end{pmatrix}$$

## 3 Proposed method

An alternative method to calculate the gradient of the activation function with respect to its inputs is to use small-signal modulation in combination with a lock-in amplifier. This method is applicable to any nonlinear material that has a known response to the input power as well as being applicable to both free space and integrated platforms. Although the method is not implemented optically as the pump probe method, it is still an analog method and so it can be implemented at high speeds besides being agnostic to the optical nonlinear material. A lock-in amplifier is a device that uses a reference tone (of a given frequency) to extract the amplitude and phase of a device (in our case, a nonlinear device/system).

It provides resilience to noise and device drift by relying on coherent mixing of the device output signal and the reference tone, i.e. coherent mixing gain is able to extract the reference signal even from a noisy and/or distorted output. Finally, the lock-in response is passed through a low pass that acts as a temporal integrator in which the product oscillates in time with an average value of zero [9]. The output is given by:

$$V_{out} = V_{sig}V_L sin(\omega_r t + \theta_{sig})sin(\omega_L t + \theta_{ref}) \tag{8}$$

Where $V_{sig}$ is the signal amplitude, $V_L$ is the lock-in amplifier output, $\omega_r$ is the reference frequency, $\theta_{sig}$ is the phase of the signal, $\omega_L$ is the lock-in amplifier frequency and $\theta_{ref}$ is the phase of the reference signal. Using the sin product identity, the output is given by:

$$V_{out} = \frac{V_{sig}V_L}{2}cos(\omega_r t + \theta_{sig} - \omega_L t - \theta_{ref}) - \frac{V_{sig}V_L}{2}cos(\omega_r t + \theta_{sig} + \omega_L t + \theta_{ref}) \quad (9)$$

This results in 2 AC components at the sum and difference frequencies of the reference and input signals. By lowpassing the output, the difference DC component is extracted with $\omega_r = \omega_L$.

$$V_{out} = \frac{V_{sig}V_L}{2}cos(\theta_{sig} - \theta_{ref}) = X \quad (10)$$

This is the signal magnitude and useful for calculating the magnitude of the derivative of the activation function with respect to the input. Essentially these steps remove the DC component of a signal. On first glance, a lock-in amplifier proves to be unnecessarily complex as the magnitude of the derivative can be simply calculated with a photodetector passed through a capacitor. The capacitor removes any DC component of the signal and the output is proportional to the modulation depth. A lock-in amplifier requires additional components besides just a photodiode, but gives larger dynamic range or sensitivity and resilience to device drift. Additionally, where the lock-in amplifier is useful is the ability to extract the phase of the signal as in ONNs the phase is a crucial component for encoding negative and complex weights. The output is dependant of the reference phase $\theta_{ref}$ and to extract the phase of the signal, the reference phase is varied and the output is measured. If the phase difference between the reference and the signal is $\pi/2$, the output is zero according to equation 8. By adding a separate circuit that multiplies the signal with a reference signal that is $\pi/2$ out of phase with the original reference signal, the low-pass output will be given by.

$$V_{out} = \frac{V_{sig}V_L}{2}sin(\theta_{sig} - \theta_{ref}) = Y \quad (11)$$

This gives 2 quantities, the $X$ and $Y$ components of the signal where the $X$ component is referred to as the in phase component and the $Y$ component is referred to as the quadrature component. The magnitude of the signal vector removes the phase dependancy:

$$R = \sqrt{X^2 + Y^2} \quad (12)$$

and the phase of the signal vector is given by:

$$\phi = tan^{-1}\left(\frac{Y}{X}\right) \quad (13)$$

This measurement can be implemented with analog electronics and therefore has the potential to operate at very high speeds [2] [10].

By mixing a small modulation with the input signal which is used for the forward pass, and tapping off the signal after the nonlinear element the gradient of the activation function with respect to the input can be calculated. An illustration of two input signals with varying power passing through the nonlinear response of a SOA is shown in Figure 1.

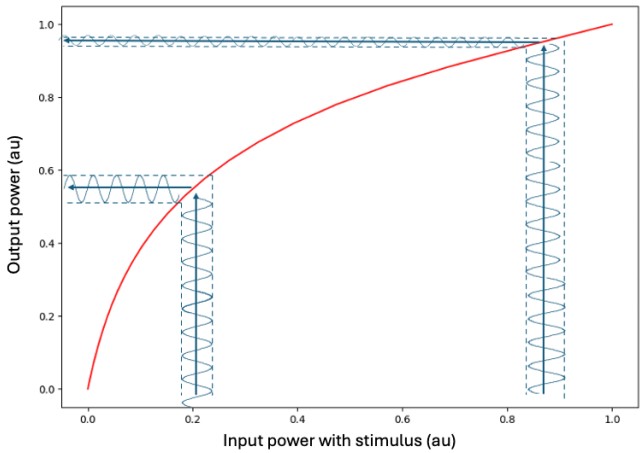

Figure 1: *Effect of a non-linear response on modulated inputs.*

Both input signals are modulated with the same small amplitude signal. As they address a different part of the nonlinear response curve, the response stimulus amplitude is different. After removing the DC component, the response to the small-signal modulation gives the gradient of the activation function with respect to the input. If the modulation depth is too high, the response can become distorted and not proportional to the gradient. When building an ONN with this technique it is essential that the small-signal modulation has a constant modulation depth.

## 4   Experiment

As a proof-of-concept we used off the shelf components, mostly in bench-top form factor, nevertheless the proposed methods are suitable for opto-electronic integration. We use a Thorlabs SOA 1013SXS as a nonlinear element. The laser is tuned to 1550 nm sent to an amplifier and is split into the pump input signal and the probe modulated signal. The pump signal passes through a digitally controllable variable optical attenuator, and the power is varied from -23 to 0 dBm. In essence, the attenuator acts as the weight matrix element. The probe signal is modulated with a 10 kHz square wave using an acousto-optical modulator (AOM), which is generated by a HP function generator with a 0.6V amplitude. The signals are then mixed using a 50/50 coupler and passed through the SOA. A photo detector (PD1) is implemented after the coupler to measure the input power. After the SOA, the signal is measured using a photo detector (PD2) and passed through to the lock-in amplifier. We use a Stanford Research Systems SR830 lock-in amplifier which is referenced to the 10 kHz square wave. The schematic of the setup is shown in Figure 2.

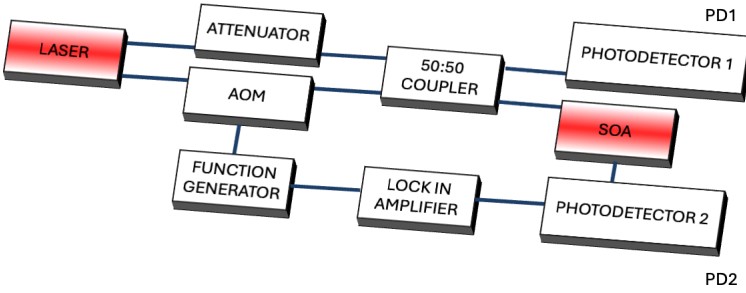

Figure 2: *Experimental setup of simultaneous gradient calculation.*

To get a comparison of the gradient measured by the lock-in amplifier with the actual gradient, the input power and output powers are measured. A fitted function based on a polynomial approximation using the steady state model of the SOA (1) is then applied to the power out as a function of the

input power to calculate an analytic gradient. An additional measure is obtained by using the numpy gradient function of the same curve. The function uses a multi-point method to approximate the derivative. To simplify the comparison between the gradient calculation methods, data plotting in Figure 3 is done with rescaling between 0 and 1. In this way, when comparing left and right plots in Figure 3, we see that at the input power range of 0.5 to 1 the gradient reaches a constant value but is clearly non-zero. In practice, rescaling would also be used due to the necessary recalibration of the system for the specific response of different nonlinear materials.

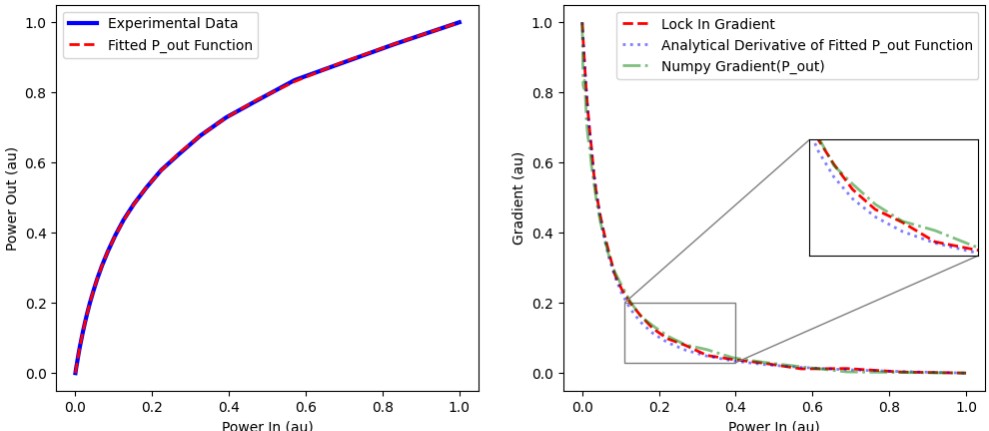

Figure 3: *Left. Nonlinear response of SOA with analytic fitted function. Right. Derivative of nonlinear response. Red: Lock-in amplifier measurement. Blue: Analytic gradient. Green: numpy gradient.*

In general, there is good agreement between the lock-in amplifier measurement and the analytic gradient. There is a slight discrepancy where the lock-in amplifier measurement is higher than the analytic gradient. Using the numpy gradient, a similar trend is observed likely indicating a slight under fitting of the analytic function. This could be due to fluctuations of the input power coming from the amplified laser or of fluctuations of the SOA output power due to environmental perturbations where the analytic function is not able to fit to this behavior.

To incorporate this technique into a full ONN, the extracted gradient or voltage can be then used to modulate a separate optical element such as a SLM or MZI mesh in which the backpropagated signal is passed through – although different schemes should be explored as well.

# 5    Conclusion

In this study, we proposed a new technique for analog gradient calculation applicable to optical neural networks and compatible with any optical nonlinear materials as well as any photonic integrated or bulk platforms. Critically, this method is applicable to networks using negative and complex weights, since the lock-in amplifier can extract the phase of the signal. And, by enabling gradient calculation synchronously with the forward pass, this method is resilient to device drift induced by environment or ageing. In summary, this technique provides a step forward in training optical neural networks, reducing the need for digital backpropagation and allowing for scalable integration of larger neural networks and faster training times.

## Acknowledgments and Disclosure of Funding

This work was supported by the UKRI Future Leaders Fellowship under Grant MR/T041218/1 and MR/Y034260/1.

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
