# OpenReview forum: "Analog Gradient Calculation of Optical Activation Function Material"
_NeurIPS.cc/2024/Workshop/MLNCP — MLNCP Poster_

### Official Review · Reviewer_rmvF · 2024-09-19
**Interesting physics, but a bigger picture view of the system would be helpful**

**Rating:** 7
**Confidence:** 3

**Review:**

The paper proposes a way to compute the derivative of the activation function with respect to its input in an artificial neural network implemented with optical physical devices. It uses a modulation of the input to read out the derivative.

I think the paper is a good fit for the workshop. However despite the ability to compute the derivative of the activation function, it is not really clear how it fits into the bigger picture of a functioning ML system. I think it is fine because the paper is more heavy on the physics side.

Therefore my main comment would be that estimating the derivative of the activation function is one relatively small part in the overall picture of neural network training. More details about how the rest would be handled by this system would help the reader better understand the potential of the approach. For instance, the derivative of the activation function typically needs to be stored and applied later on the backpropagated error vector at suitable times. How does the proposed system handles the backpropagation of the error? Is it relying on some other algorithm?

Minor comment: the example of applying ReLU to a weight matrix is strange because the activation function is not applied on the weights but on the matrix vector product of the weight and previous layer.

Question: Figure 3 is nice but if the right plot is the derivative of the left plot, then why does the slope at zero on the left seems greater than one, which is the value on the right plot at zero?

---

### Official Review · Reviewer_dU2s · 2024-10-03
**A novel method for estimating the weight gradients in ONNs**

**Rating:** 6
**Confidence:** 2

**Review:**

The manuscript proposes a novel implementation of backpropagation (BP) in optical neural networks (ONNs). The method performs BP in the analog domain. To achieve this, the idea explored in this work relies on an analog opto-electronic method that uses small-signal modulation and a lock-in amplifier.



**Strengths**

The method is applicable to networks with negative weights and complex-valued weights, using the lock-in amplifier to extract the signal’s phase.

The authors find a good agreement between the measured gradient (lock-in amplifier measurement), the analytic gradient, and the numpy gradient.

Alternative implementations of BP in ONNs are discussed. For example, a prior method by Pai et al. relied on calculating nonlinear activation functions and their derivatives in the digital domain. Another prior method by Spall et al. implemented the activation function and its derivative in the analog domain, using a pump (for forward signals) and a probe (for backward gradients) in rubidium gas, but probe gradient measurement is accurate only for a narrow range of pump powers. Advantages of the proposed method with respect to finite difference-based methods are also discussed.

Note: I cannot judge the advantages and drawbacks of the approach compared to existing methods, or the claims about speed and energy-efficient of the approach.



**Remarks**

References did not compile correctly.

Some typos:
Line 34: used
Line 45: were -> where
Line 153: fitted function fitted function

---

### Decision · Program_Chairs · 2024-10-10

Accept (Poster)